# Effects of the Severity of Wildfires on Some Physical-Chemical Soil Properties in a Humid Montane Scrublands Ecosystem in Southern Ecuador

**Vinicio Carrión-Paladines** [1,*] , **María Belén Hinojosa** [2], **Leticia Jiménez Álvarez** [1] , **Fabián Reyes-Bueno** [1] , **Liliana Correa Quezada** [3] and **Roberto García-Ruiz** [4]

1 Departamento de Ciencias Biológicas y Agropecuarias, Universidad Técnica Particular de Loja, San Cayetano Alto s/n, Loja 1101608, Ecuador; lsjimenez@utpl.edu.ec (L.J.Á.); frreyes@utpl.edu.ec (F.R.-B.)
2 Departamento de Ciencias Ambientales, Universidad de Castilla-La Mancha, Campus Fábricas de Armas, 45071 Toledo, Spain; mariabelen.hinojosa@uclm.es
3 Departamento de Ciencias Jurídicas, Universidad Técnica Particular de Loja, San Cayetano Alto s/n, Loja 1101608, Ecuador; ldcorrea@utpl.edu.ec
4 Departamento de Biología Animal, Biología Vegetal y Ecología, Sección de Ecología, Universidad de Jaén, España, Campus Las Lagunillas, Edificio Ciencias Experimentales y de la Salud (B3), 23071 Jaén, Spain; rgarcia@ujaen.es
* Correspondence: hvcarrionx@utpl.edu.ec

**Abstract:** Humid montane scrublands (HMs) represent one of the least studied ecosystems in Ecuador, which in the last decade have been seriously threatened by the increase in wildfires. Our main objective was to evaluate the effects of wildfire severity on physicochemical soil properties in the HMs of southern Ecuador. For this purpose, fire severity was measured using the Normalized Burn Ratio (NBR) and the difference between pre-fire and post-fire (NBR Pre-fire-NBR Post-fire) over three contrasted periods (years 2019, 2017, and 2015) was determined. Likewise, 72 soil samples from burned HMs and 72 soil samples from unburned HMs were collected at a depth of 0 to 10 cm, and some physical (bulk density and texture) and biochemical (pH, soil organic matter, and total nutrients) soil properties were analyzed and statistically processed by one-way ANOVA and principal component analysis (PCA). Results indicate that burned HMs showed mixed-severity burning patterns and that in the most recent wildfires that are of high severity, SOM, N, P, Cu, and Zn contents decreased drastically (PCA: component 1); in addition, there was an increase in soil compaction (PCA: component 2). However, in older wildfires, total SOM, N, P, K, and soil pH content increases with time compared even to HMs that never burned (*p*-value < 0.05). These results can help decision makers in the design of policies, regulations, and proposals for the environmental restoration of HMs in southern Ecuador affected by wildfires.

**Keywords:** humid montane scrublands; wildfire severity; soil properties

## 1. Introduction

Wildfires are a major environmental problem in many forest biomes around the world [1]. They are often caused by lightning [2,3], but humans also have a profound effect on fire regimes through the introduction of ignition sources [4]. According to Benali et al. [5], human activity is responsible for lighting the majority of all fires. In this context, recent research suggests that a warming climate and increased human ignition could increase the frequency of wildfires around the world [6,7]. Quantifying future global wildfire activity is, therefore, challenging due to uncertainties among land cover trends and in fire-climate relationships [7]. This is also the case in Latin America, where most wildfires are of anthropogenic origin where slash-and-burn is practiced [8]. Many villages use fire to make land-use changes and convert forests to cropland and pasture [9]. Although slash-and-burn is a very old practice, several fires often get out of control when climatic factors and fuel

availability are not taken into account, resulting in large fires [10]. Whatever their origin, changes in the wildfire regime are disrupting the health of ecosystems, especially those considered to be sensitive [11].

The main impact of wildfires is the destruction of plant cover, which causes a loss of biodiversity and accelerates soil erosion processes. Bodí et al. [12] reported that the degree of severity of a wildfire, understood as the level of impact on an ecosystem, is generally different when they occur in the same ecosystem and even within the same fire. This depends on fire weather, which can change abruptly, water balance, topography (slope), vegetation type, fuel load, and fuel moisture content [13–15]. According to Ayoubi et al. [16], these processes can greatly alter the physical and chemical characteristics of the soil, causing a resultant loss of nutrients. As temperature increases during a fire, the temperature above and below the ground also increases, which causes heating of the mineral content and the consumption of organic matter (SOM) [17], causing changes in bulk density, porosity, texture, color, moisture content, and effects on permeability [18,19]. However, after a wildfire, the effects and/or changes can be positive or negative on the ground. This depends on the heat intensity (Kcal/kg) of the fire, resulting in fires of low or high severity. According to Keeley et al. [20], low severity fires are part of the natural dynamics of most ecosystems on earth. Many plant communities are adapted to low-severity wildfires, as they have developed fire-induced germination mechanisms and zero tolerance [21], where changes in soil properties are usually ephemeral [22]. The main changes that occur in soils of low-severity wildfires are decreases in microbial respiration and enzyme activity [23], as well as to increases in soil pH and soil organic matter concentration (SOM). According to Chandra and Bhardwaj [18], low severity fires cause the combustion of SOM but allow an increase in the availability of nutrients, which favors the regeneration of grass and the growth of the plant community after the fire. Likewise, Sulwiński et al. [24] found that in moderately burned areas (low severity), relatively high phosphate contents are recorded. In contrast, when a fire is of high severity, it causes a complete loss of SOM and volatilization losses of nitrogen, available phosphorus, and potassium, and it has been shown that a very high temperature is required for the complete combustion of Mn, Mg, Cu, and other nutrients [25]. In addition, it can also cause an alteration to the stability of soil aggregates [26], which can cause, depending on climatic conditions, changes to topography, vegetation, soil type, and texture, leading to erosion and runoff after wildfires [27].

On the other hand, in ecosystems where wildfires are historically frequent, the biogeochemical properties of vegetation and soil often return to pre-fire conditions over years or decades [28]. Therefore, the natural recovery process of soil quality and health after a fire is very slow. For example, Choromanska and DeLuca [29] showed that C and N contents decreased after a wildfire and did not recover after 9 months of study in a pine (*Pinus ponderosa*) and fir (*Pseudotsuga menziesii*) forest in the United States. Likewise, for humid tropical forests, some researchers recommend 4 years for the recovery of C and N contents [30]. As such, more research is needed in this field, taking into consideration the different types of ecosystems, environments, and especially the degree of severity of the fire.

In South America, in terms of areas burned/year, the most affected countries are Brazil and Bolivia (4% of their territories) while the countries with the highest number of active fires/unit areas are Guatemala, Paraguay, and Honduras [31]. Ecuador has also suffered an increase in the frequency, intensification, and extent of wildfires in recent decades. According to Armenteras et al. [32], in Ecuador, wildfires occur due to the great variety of geographic regions, different ecological floors, as well as climatic variability, where different human groups such as mestizos, indigenous and afro-descendants live in differing economic conditions. In Ecuador, mestizo and indigenous ethnic groups continue to use traditional slash-and-burn as a strategy for food production [33]. After having applied fire to their fields and/or orchards, they sow maize (*Zea mays*) and bean (*Phaseolus vulgaris*) seeds among other crops [34] upon a layer of ash bed in the topsoil. However, these areas are located at the edge of forests so that, at the slightest carelessness, if the necessary

measures are not observed, the fire advances towards the forest producing wildfires that are of great magnitude and intensity [10]. Despite the fact that wildfires are becoming very recurrent in Ecuador [35], there is very little research on the effects they have on ecosystems. In general, most studies were developed to establish prevention programs [36], determine the social perception of the effects of fires on sustainable tourism [37], and studies of fire ecology in the high tropical Andes [38]. However, only Suárez and Medina [39] evaluated the impact of wildfires on vegetation structure and soil properties in the paramos of northern Ecuador. In addition, in Ecuador, laws exist that punish with imprisonment of three to six months in jail [40] for those carrying out agricultural burning that becomes uncontrollable and causes wildfires. These extreme measures are in lieu of forest control in Ecuador, which is inadequate and creates difficulty in the identification and sanctioning of those who cause wildfires, especially arsonists, who are often not sanctioned at all [41]. In this context, it is necessary to generate new scientific information where methods are applied to determine the severity of the fires (remote sensing) and to know the effects on the physical and chemical properties of the soils. In this way, it will be possible to elucidate this problem, highlighting the fact that this research can be applied throughout the country [42].

One of the ecosystems most affected by wildfires in Ecuador is montane humid scrublands (HMs). HMs are characterized as being bushy vegetation of small native species (up to 6 m) that are covered with epiphytes and liverworts [43,44]. They are distributed in the northern, central, and southern Andes of Ecuador [45,46] where there are endemic species that are in danger of extinction, especially in the Tumbes region (e.g., *Artibeus fraterculus* [47]). HMs altitudinally are distributed in Ecuador between 2000 and 3000 m.a.s.l [46,48]. The floristic composition of these thickets varies according to location, humidity, and soil type, and many of them have multiple ethnobotanical applications [43]. According to Torracchi et al. [49], these ecological formations are in a critical situation being that the dynamics of deforestation have brought to the brink their total disappearance, with a destruction rate of 0.44% per year. In southern Ecuador, they are the most representative plant units of shrubby vegetation, which do not have a defined stem and maintain the greenness of its leaves constantly throughout the year [44]. In these HMs, 78 species have been found, of which 20 are endemic to the Tumbesian region and are mainly represented by species such as the cucharillo (*Oreocallis grandiflora* (Lam.) R.Br.), chinchas of the genus *Chusquea* sp., huaycundos or bromeliads of the genera *Guzmania* sp., *Tilladsia* sp., and *Puya* sp., cashco (*Weinmannia glabra* Lf), dumarín (*Tibouchina laxa*), and asteraceae of the genus *Baccharis* sp., among others [44,50]. In addition, these ecological formations play an important role in the management of the micro-watersheds that supply drinking water to the populations of southern Ecuador [51]. According to Balcázar and Reyes [52], in the south of the country, wildfires among HMs are caused by human activities through the implementation of agricultural burning; however, no studies have been conducted on the impacts of wildfires on this ecosystem, which would provide insight into the behavior of fire and the effects on soil ecology. In this context, studies are required to cover this research gap.

The objective of this study was to determine the effects of wildfires on some physical and chemical parameters of soil quality in the HMs of southern Ecuador, since these ecosystems are vulnerable to increasing anthropogenic pressures, due to the change of land use to promote agricultural activities and the implementation of rural and urban infrastructure [52,53]. For this purpose, soil sampling was conducted in areas of HMs that had the same topographic, vegetation, altitudinal, and climate characteristics and where wildfires occurred, during a small chronosequence of three contrasting periods, corresponding to the years 2019, 2017, and 2015. The analysis includes the study of the wildfire history utilizing data and georeferenced records, the determination of the severity of the fires, and the laboratory analysis of soil properties. This information could help decision-makers to understand the impact of wildfires on HMs, and with this, appropriate policies and regulations could be implemented for the sustainable environmental management of this ecosystem in southern Ecuador.

## 2. Materials and Methods

### 2.1. Study Area

The study area was located in the Loja canton (−79.096 to −79.543 and −3.666 to −4.511) where elevation varies from 2100 to 3420 m asl (Figure 1). Four subclimates characterize the canton: cold, subtropical, tropical, and temperate. The latter occupies the widest range in the canton [54]. The mean annual temperature varies between 15.3 °C in the valleys and 7.3 °C in the mountain ranges [55]. Annual rainfall ranges from 500 to 2000 mm per year [56]. There are two well-defined climatic periods during the year: a dry period during September and October [57,58] and a wet period from December to April when highest maxima monthly rainfall is recorded. During the rainy season, there are many high-intensity rainfall events that occur due to the storms that pass over the inter-Andean valleys [55].

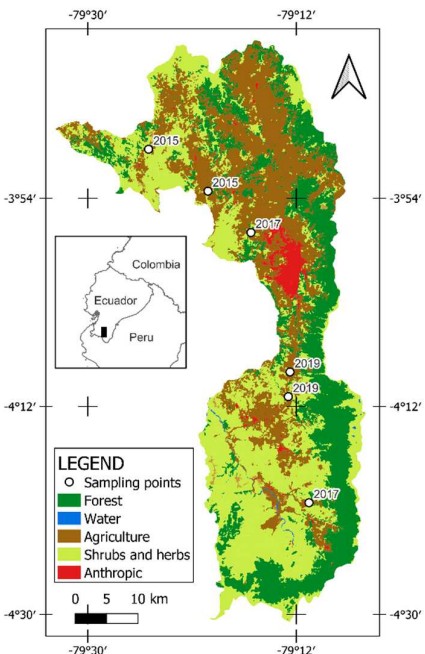

**Figure 1.** Location of the study area, Loja canton, southern Ecuador.

The Loja canton has a great diversity of ecosystems which range from the southern montane evergreen forests (eastern mountain range of the Andes) and the humid montane scrublands (HMs) [59,60]. The main plant species of the humid montane scrublands (HMs) include *Oreocallis grandiflora* (Lam.) R.Br., *Freziera verrucosa* (Hieron.) Kobuski, *Baccharis latifolia* (Ruiz & Pavón) Pers., *Mutisia magnifica* C. Ulloa & P. Jørg., *Cleome longifolia* Willd. Ex Schult., *Elaphoglossum* sp. Schott ex J. Sm., *Tibouchina laxa* (Desr.) Cogn., *Paspalum humboldtianum* Flüggé, *Brugmansia arborea* (L.) Steud, *Cestrum tomentosum* Moc. & Sessé ex Dunal, *Streptosolen jamesonii* (Benth.) Miers, *Passiflora ligularis* Juss., and *Myrsine sodiroana* (Mez) [44,59]. The predominant soils in HMs are Entisols and Inceptisols [56,61].

### 2.2. Study Design

The study was developed in three fundamental methodological steps. The first consisted of the identification of wildfires in HMs during three contrasting years (2019, 2017, and 2015) using remote sensing methods, the Visible Infrared Imaging Radiometer, Moderate Resolution Imaging Spectroradiometer (MODIS sensor), and NASA fire information. The second step consisted of the determination of fire weather with NASA data that allowed the elaboration of the respective climographs and the determination of the severity levels of the fires through remote sensing methods using Sentinel 2 satellite images. The final step comprised determining the effect of wildfires on the physical and chemical properties of

the soil by using laboratory analysis. The following is an outline of the workflow, referring to the methodology applied in this study (Figure 2).

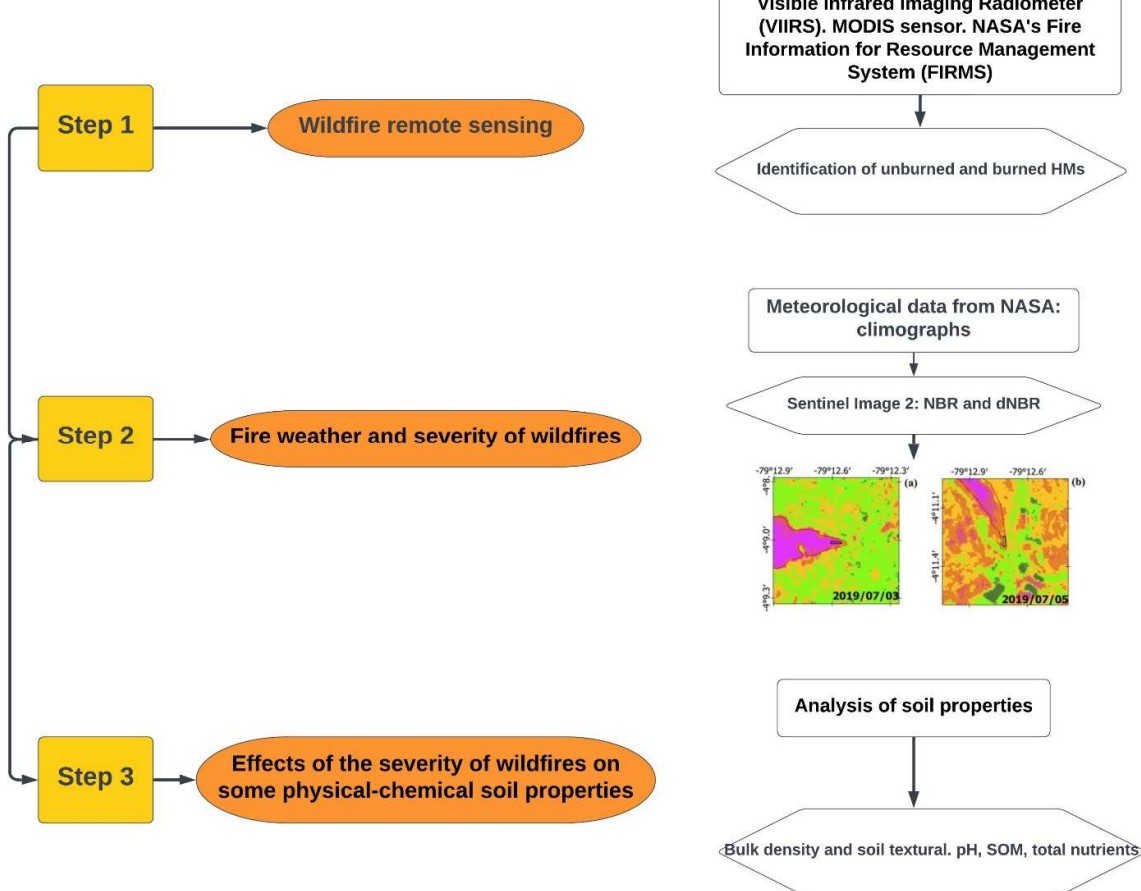

**Figure 2.** Schematic workflow of the overall study methodology. For the three steps of this study, the various data flows and analytical steps are shown, as well as how these steps are interrelated.

### 2.3. Identification and Description of HMs Affected by Wildfires

In order to identify the perimeters of wildfires affecting HMs, we used fire data derived from the 375 m and 750 m VIIRS (VIIRS 375 m NRT (NOAA-20), NASA_FIRMS [62]) and MODIS sensor-derived heat source data (NASA_FIRMS [63]). In addition, we used wildfire records from the database of the National Risk and Emergency Management Service of Ecuador [64] covering the period 2015–2019. These data were loaded into the open-source software QGIS 3.12.2-București [65], and we selected two HM sites burned in 2019, two HMs burned in 2017, and two HMs burned in 2015 with comparable pre-fire vegetation, topography, and soil type. For each of them, a nearby and comparable HM (in terms of soil type, topographic, and landscape characteristics) but in a natural state (which was not affected by any wildfire) was also selected.

### 2.4. Determination of Fire Weather and Severity of Wildfires

The fire weather was determined using meteorological data from NASA (https://power.larc.nasa.gov/data-access-viewer/ accessed on 5 November 2021) [66] corresponding to temperature (°C), precipitation (mm), relative humidity (%), and wind speed (m/s converted to km/h) in each of the contrasting burned areas, considering each year of the fire. With the data, the respective annual climographs were generated that demonstrated fire behavior and also allowed for interpretation of the fire severity indices (Figure 3).

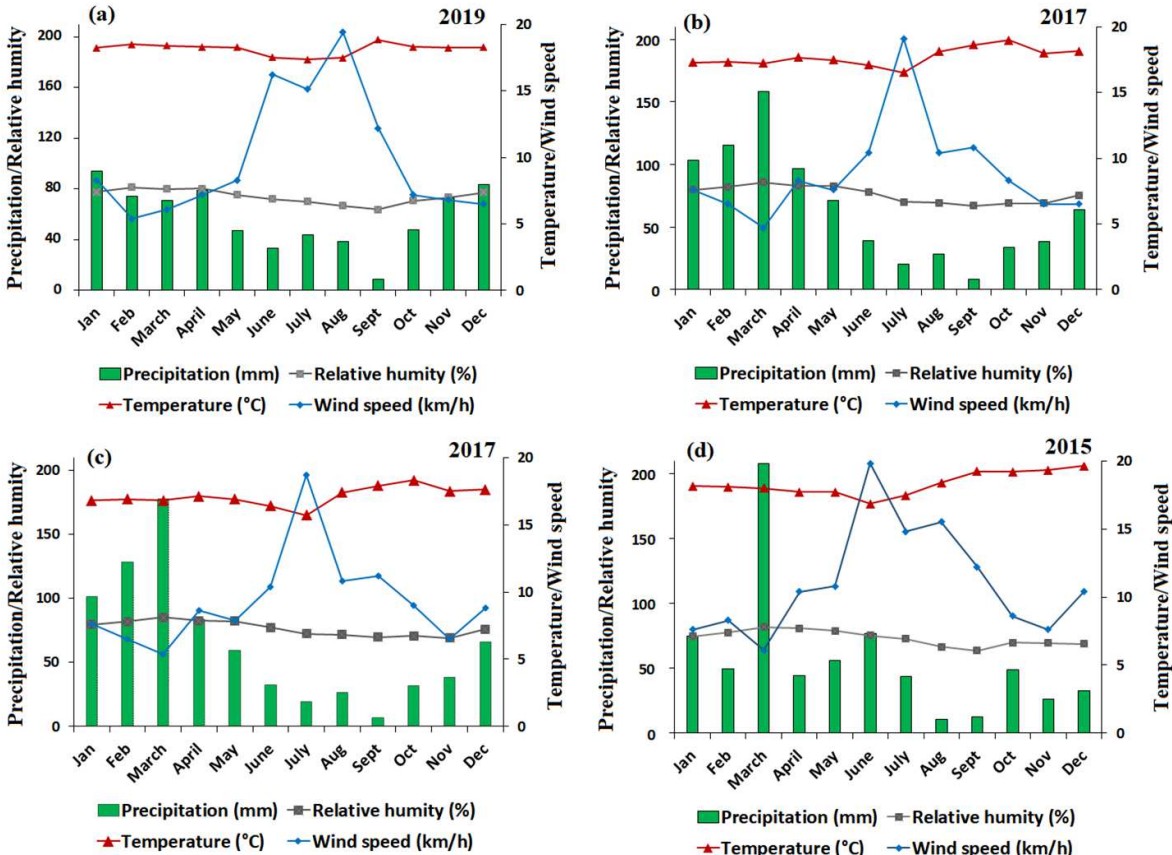

**Figure 3.** Climographs according to each sampling site. (**a**) Climograph of zones 1 and 2 of the year 2019. (**b**) Climograph of zone 1 of the year 2017. (**c**) Climograph of zone 2 of the year 2017. (**d**) Climograph of zones 1 and 2 of the year 2015. The climatic data were determined using meteorological data from NASA.

For the 2019 wildfires, one climograph was obtained, while two were generated for the 2017 wildfires and one for the 2015 wildfires (Figure 3). Only one climograph was generated for burned areas of close proximity (3.9 km) to each other (2019 and 2015 fires), where the NASA predictive model data collected the same values for each meteorological variable. However, for the 2017 fires, it was possible to construct two climographs, because the two events are far apart (44.4 km); therefore, the NASA predictive model data collect different values for each meteorological variable (Figure 3b,c). In the HMs of southern Ecuador, the favorable conditions for wildfire occurrence are the months of August and September, when precipitation decreases (26.0 and 9.3 mm on average respectively), relative humidity decreases (68.6 and 66.0%, respectively), and there is an increase in wind speed (14.0 and 11.6 km/h, respectively) and temperature (17.8 and 18.6 °C, respectively) compared to the other months of the year.

For the analysis of wildfire severity, Sentinel 2 Level 1C satellite images from the years 2017 and 2019 with the MSI multispectral sensor were used; however, the image from the year 2015 could not be obtained since such images in Sentinel are active as of 2016. In this context, we tried to use a cloud-free Landsat image from a date after the 2015 fires, which could only be identified 8 to 10 months after the fire, after the rainy period (austral winter) so it was not useful to determine the severity of the wildfires. In addition, we considered the dates with the lowest cloud cover (<20% cloud cover), which corresponded to the last months of the year, where the highest incidence of wildfires occurs [67]. Normalized Burn Ratio (NBR) allowed us to highlight burned areas by their spectral signature [68]. The analysis was performed using the following formula [69].

$$NBR = \frac{(R_{NIR} - R_{SWIR})}{(R_{NIR} + R_{SWIR})}$$
$$R_{NIR} = reflectivity\ in\ the\ band\ NIR\ (B8A)$$
$$R_{SWIR} = reflectivity\ in\ the\ band\ SWIR\ (B12)$$

(1)

Likewise, dNBR, the difference between pre-fire and post-fire NBR (NBR Prefire-NBR Postfire) was calculated to estimate the severity using the respective formula [70] (Figure 4).

$$dNBR = (NBR_1 - NBR_2)$$
$$NBR_1 = pre - fire\ burned\ area\ index$$
$$NBR_2 = post - fire\ burned\ area\ index$$

(2)

In the HMs studied, there is a wide typology of severities ranging from high severity (Figure 4a,d), low severity (Figure 4b), and moderate-low severity (Figure 4c). Thus, for the year 2019, zone 1 presents a high severity and zone 2 a low severity; for the 2017 fires, zone 1 of the burned HMs presents a moderate-low severity while zone 2 presents a high severity. Severity maps for 2015 were not obtained for the reasons of lack of satellite information from both Sentinel 2 Level 1C and Landsat image reported above.

Both climographs and severity maps are different due to the distances of the plots (Figure 1) and, therefore, the NASA predictive model data collected different values for each meteorological variable and the remote sensing of fire severity also presented a different response (Figures 3 and 4). In this context, Table 1 shows additional characteristics of the HMs where wildfires occurred with their respective unburned controls. The sites are characterized by slopes greater than 70%, similar textures, the same pre-fire vegetation type, and altitudes (from 1660 to 2658 m asl) within the range for HMs inhabiting the inter-Andean alley [43,60].

*2.5. Soil Sampling and Laboratory Analysis*

After determining the severity of the wildfires in each of the burned HMs, the areas with the largest area affected by each type or level of severity were identified. Once the areas with greater size-severity were identified, accessibility was considered for soil sampling (Figure 4). Thus, in each of the identified sites, under burned and unburned treatments, three 20 m × 20 m plots were installed (400 m$^2$ each, 2400 m$^2$ per site, 14,400 m$^2$ in total). Each plot was evenly spaced at least 100 m apart. Soil samples were taken at each corner of each plot at a depth of 0–10 cm, using standardized metal cores (6 cm diameter, 10 cm height, 283 cm$^3$ volume) [71]. The 0–10 cm depth was considered because generally, at this depth, fire increases soil temperature and affects the main physical and chemical properties, as reported in previous research [72,73]. In each plot, 4 samples were taken separately for further analysis of bulk density, texture, pH, soil organic matter (SOM), and total soil nutrients (3 years × 2 study sites × 2 treatments × 3 plots × 4 samples = 144 individual samples). Once the soil samples were collected, they were packaged in separately labeled polypropylene bags.

Soil bulk density was determined after drying the soil of the cylinders in an oven for 48 h at 105 °C. Soil texture was determined using the bouyoucos hydrometer method [70,74]. The pH was measured with a pH meter applying the standard method [71,75]. SOM was determined using the Walkley and Black method [72,76]. The total nitrogen (TN) of the soil was determined by the Kjeldahl method. The available phosphorus content was determined by the modified Olsen method [73,77]. Total potassium (cmol/kg), total calcium (cmol/kg), total magnesium (cmol/kg), total manganese (mg/kg), total copper (mg/kg), and total zinc (mg/kg) were determined by the atomic absorption spectrophotometry method [78].

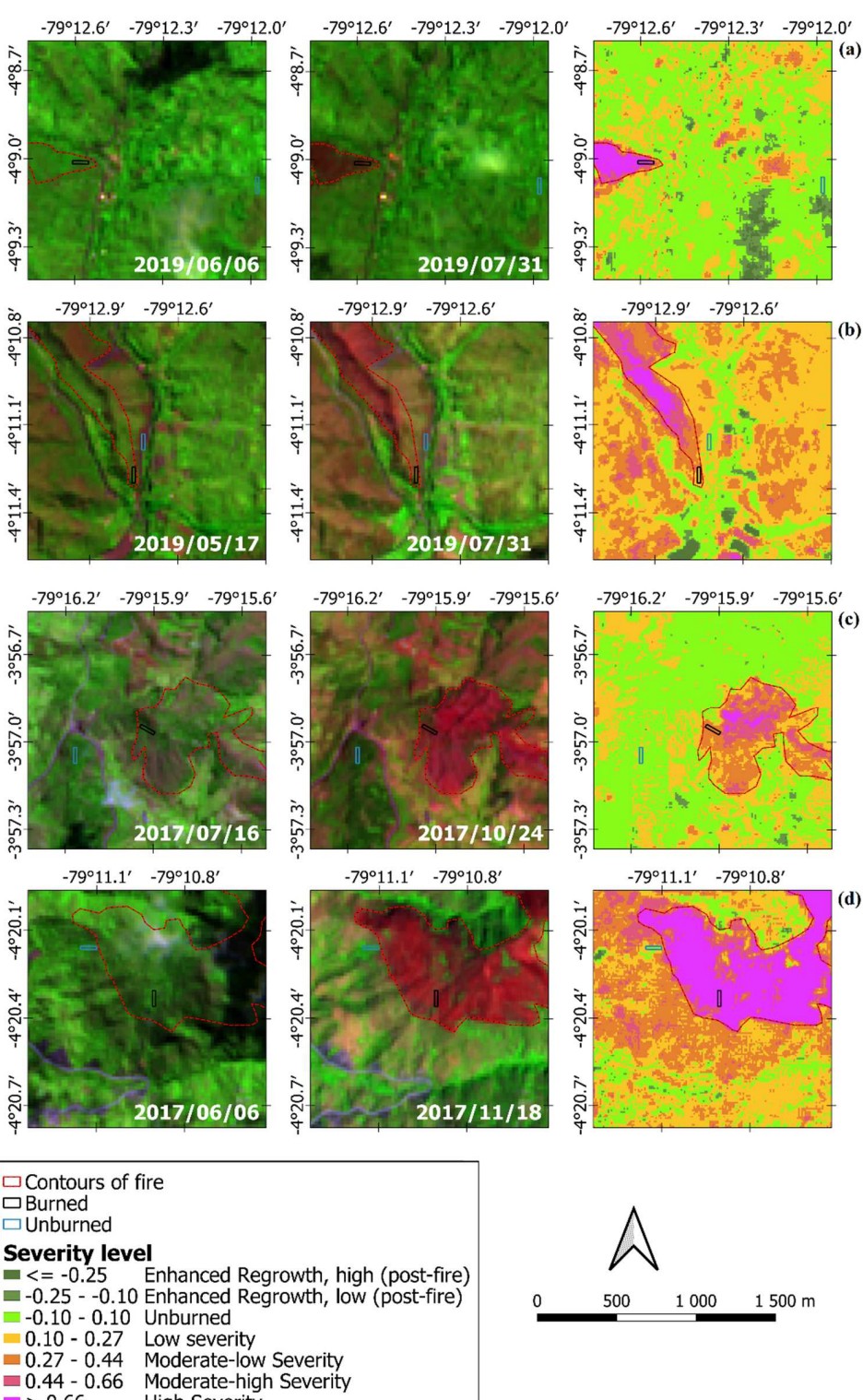

**Figure 4.** Severity maps according to each sampling site. Scenarios before, after, and severity of the wildfire for burned HMs are presented. In addition, the location of the soil sampling plots for both burned and unburned HMs are presented. (**a**) High severity map—year 2019. (**b**) Low severity map—year 2019. (**c**) Moderate low severity—year 2017. (**d**) High severity map–year 2017. The severity map was determined using normalized burn ratio (NBR) and the difference between pre-fire and post-fire (NBR Prefire–NBR Postfire).

**Table 1.** Main characteristics of the studied sites with HMs of Loja canton, southern Ecuador.

| Study Site/Year | Time Since Fire (Years) | Fire Occurrence | Plot Code | Coordinates | Altitude (m) | Vegetation Cover | Slope of the Land |
|---|---|---|---|---|---|---|---|
| 1/2019 | 2 | Unburned | 1UB | 4° 9′5.62″ S 79°11′42.31″ W | 1997 | 75% shrub, 20% grass, 5% tree. | >70% |
| | | Burned high severity | 1B-HS | 4° 9′1.76″ S 79°12′34.93″ W | 1822 | | |
| 2/2019 | 2 | Unburned | 2UB | 4°11′17.91″ S 79°12′43.37″ W | 1665 | 50% shrub, 35% grass, 15% tree. | >70% |
| | | Burned low severity | 2B-LS | 4°11′9.46″ S 79°12′41.24″ W | 1660 | | |
| 3/2017 | 4 | Unburned | 3UB | 4°20′9.58″ S 79°11′7.66″ W | 2066 | 60% shrub, 30% grass, 10% tree. | >70% |
| | | Burned | 3B-MLS | 4°20′19.90″ S 79°10′54.31″ W | 2220 | | |
| 4/2017 | 4 | Unburned | 4UB | 3°57′2.91″ S 79°16′9.96″ W | 2620 | 75% shrub, 20% grass, 5% tree. | >70% |
| | | Burned high severity | 4B-HS | 3°56′55.29″ S 79°15′55.29″ W | 2658 | | |
| 5/2015 | 6 | Unburned | 5UB | 3°49′28.83″ S 79°24′50.53″ W | 2313 | 50% shrub, 25% grass, 25% tree. | >70% |
| | | Burned | 5B | 3°49′44.95″ S 79°24′45.06″ W | 2320 | | |
| 6/2015 | 6 | Unburned | 6UB | 3°53′31.97″ S 79°19′34.98″ W | 2150 | 50% shrub, 25% grass, 25% tree. | >70% |
| | | Burned | 6B | 3°53′22.65″ S 79°19′37.73″ W | 2187 | | |

*2.6. Statistical Analysis*

Differences in soil properties were tested by one-way analysis of variance (ANOVA, F test, $p < 0.05$) considering each year of study separately and comparing the different levels (different years since the fire, its controls, and fire severity levels). If the statistical analysis was significant, the means were compared with Tukey's honestly significant difference (HSD) post hoc test and accepted with a value of $p < 0.05$ in all cases. In addition, relationships between soil characteristics of burned areas were evaluated using principal component analysis (PCA) [79]. PCA was based on the standardized soil variable correlation matrix to explore the different responses of soils in distinct fire years. Analysis of variance (ANOVA) was performed using SPSS statistical software (v.15.0; SPSS Inc., Chicago, IL, USA) and the PAST program, version 3 [80], was used to obtain the principal components (PCA).

**3. Results**

*3.1. Effects of the Severity of Wildfires on the Physical Properties of the Soil in Each Contrasted Area*

Table 2 presents the results of the bulk density at a depth of 0–10 cm. This physical parameter varied between 0.8 g cm$^{-3}$ and 1.4 g cm$^{-3}$ showing significant differences per year and between unburned HMs (UB) with their respective burned HMs (B). For the year 2019, which is the most recent event, there were two types of fires, one of high severity (1B-HS) and another of low severity (2B-LS) (Figure 4a,b). In 1B-HS, there has been relative compaction of the soil (1.4 g cm$^{-3}$) with regards to 1UB (1.1 g cm$^{-3}$) while in 2B-LS (1.1 g cm$^{-3}$) compaction decreased with respect to 2UB (1.3 g cm$^{-3}$) (Table 2). This indicates that in places where wildfires have recently occurred, soil compaction depends on the severity of the fire. Likewise, for the wildfires of 2017 (older), there were fires of moderate low severity (3B-MLS) and high severity (4B-HS) where we can observe that in 3B-MLS,

the bulk density decreases relatively (3UB 0.9 g cm$^{-3}$ to 0.8 g cm$^{-3}$, respectively) and in 4B-HS, bulk density follows a similar pattern (4UB 1.2 g cm$^{-3}$ to 1.1 g cm$^{-3}$, respectively). For the 2015 wildfires (the oldest), there is a relative decrease in bulk density for the first zone and it is maintained for the second zone. In addition, the soils studied have very similar textures ranging from sandy loam to clay loam (Table 2), and as such, this physical property has not been affected by wildfires.

**Table 2.** Mean values of the physical properties of the soil for the different HMs, including their standard deviations (12 repetitions for each HM). Different letters mean significant difference among burning scenarios within each study year (*p* < 0.05, Tukey HSD).

| Year | Plot Code | Bulk Density | Sand | Silt | Clay | Textural Class |
|---|---|---|---|---|---|---|
| | | (g cm$^{-3}$) | % | % | % | |
| 2019 | 1UB | 1.1 ± 0.1 [a] | 61.6±1.0 [a] | 18.0 ± 0.0 [a] | 20.4 ± 1.0 [a] | Sandy loam |
| 2019 | 1B-HS | 1.4 ± 0.1 [c] | 72.5 ± 0.3 [b] | 14.9 ± 0.0 [a] | 12.6 ± 3.0 [b] | Sandy loam |
| 2019 | 2UB | 1.3 ± 0.2 [b] | 71.6 ± 3.0 [b] | 15.0 ± 3.0 [a] | 13.4 ± 0.0 [b] | Sandy loam |
| 2019 | 2B-LS | 1.1 ± 0.1 [a] | 69.6 ± 1.0 [b] | 16.0 ± 0.0 [a] | 14.4 ± 1.0 [b] | Sandy loam |
| 2017 | 3UB | 0.9 ± 0.2 [a] | 64.6 ± 6.0 [a] | 14.0 ± 2.0 [a] | 21.4 ± 8.0 [a] | Sandy loam |
| 2017 | 3B-MLS | 0.8 ± 0.3 [c] | 79.0 ± 3.6 [b] | 9.4 ± 3.5 [a] | 11.5 ± 0.1 [a] | Sandy loam |
| 2017 | 4UB | 1.2 ± 0.2 [b] | 72.5 ± 3.0 [a] | 16.9 ± 0.0 [a] | 10.6 ± 3.0 [a] | Sandy loam |
| 2017 | 4B-HS | 1.1 ± 0.1 [b] | 64.0 ± 0.6 [a] | 19.4 ± 1.5 [a] | 16.5 ± 0.9 [a] | Sandy loam |
| 2015 | 5UB | 1.0 ± 0.2 [a] | 64.6 ± 6.0 [a] | 14.0 ± 2.0 [a] | 21.4 ± 8.0 [a] | Sandy clay loam |
| 2015 | 5B | 0.8 ± 0.1 [b] | 74.6 ± 4.0 [b] | 10.0 ± 4.0 [a] | 15.4 ± 0.0 [a] | Sandy loam |
| 2015 | 6UB | 1.1 ± 0.1 [a] | 66.6 ± 2.0 [a] | 13.0 ± 1.0 [a] | 20.4 ± 1.0 [a] | Sandy loam |
| 2015 | 6B | 1.1 ± 0.1 [a] | 67.6 ± 1.0 [a] | 10.0 ± 0.0 [a] | 22.4 ± 1.0 [a] | Sandy clay loam |

### 3.2. Effects of Wildfires on Chemical Properties of Soils

The wildfires caused marked changes in some soil chemical properties (Figure 5) showing that, for each sampling year, in the burned HMs, there is an increase and/or a decrease in certain nutrients. These changes depended on fire severity and recovery time (natural succession), as demonstrated in other research studies [81]. In HMs, wildfires were of moderate-low severity, low severity, and high severity (Figure 4). In these scenarios, it could be observed that in the most recent fires (year 2019, 2 years after the fire), there is a decrease or increase in SOM (%) depending on whether they were of high severity (1B-HS) or low severity (2B-LS) (*p*-value < 0.05). The 1B-HS presented a decrease in SOM (from 5.5% of the 1UB control to 2.2% in the 1B-HS). In 1B-HS, in general, there is a loss of SOM and, in addition, losses due to volatilization of nitrogen (from 0.3% of the 1UB control to 0.1% in the 1B-HS), available phosphorus (from 16.3 mg/kg 1UB control to 7.5 mg/kg in 1B-HS), and potassium (from 0.4 cmol/kg 1UB control to 0.2 cmol/kg in 1B-HS); likewise, there are reductions in Mn, Mg, and Cu, as reported in recent research [25] (Figure 5). In 2019 (2B–LS), there was instead a relative increase in SOM (from 3.5% in 2UB to 3.8% in 2B–LS).

For the 2017 wildfires where there is a longer recovery time (4 years) but with fires of moderate-low (3B-MLS) and high severity (4B-HS), we observed that there is an increase in SOM from 7.5% in 3UB to 11.6% in 3B-MLS and from 6.0% in 4UB to 6.6% in 4B–HS (*p*-value < 0.05). However, for nitrogen alone, there is a decrease for 4B-HS (from 0.8% of the 4UB control to 0.3% in the 4B-HS) but an increase for available phosphorus (from 14.5 mg/kg of 3UB control to 20.2 mg/kg in 3B-MLS, and 14.7 mg/kg of 4UB control to 24.2 mg/kg in 4B-HS), as well as K, Ca, Mn, Cu, and Zn (Figure 5). In the case of older fires (2015, 6 years after the fire), although we do not know the degree of severity, the recovery time (total regeneration of soil physical-chemical properties) indicates that 6 years after we performed the edaphic analysis, the effect of fire has increased the chemical properties of the soil, even with higher contents than the previously unburned HMs (Figure 5). Furthermore, the pH range observed in Figure 5 (range of 4.5–7.2) shows that this factor is in a range considered optimal in the soil [82].

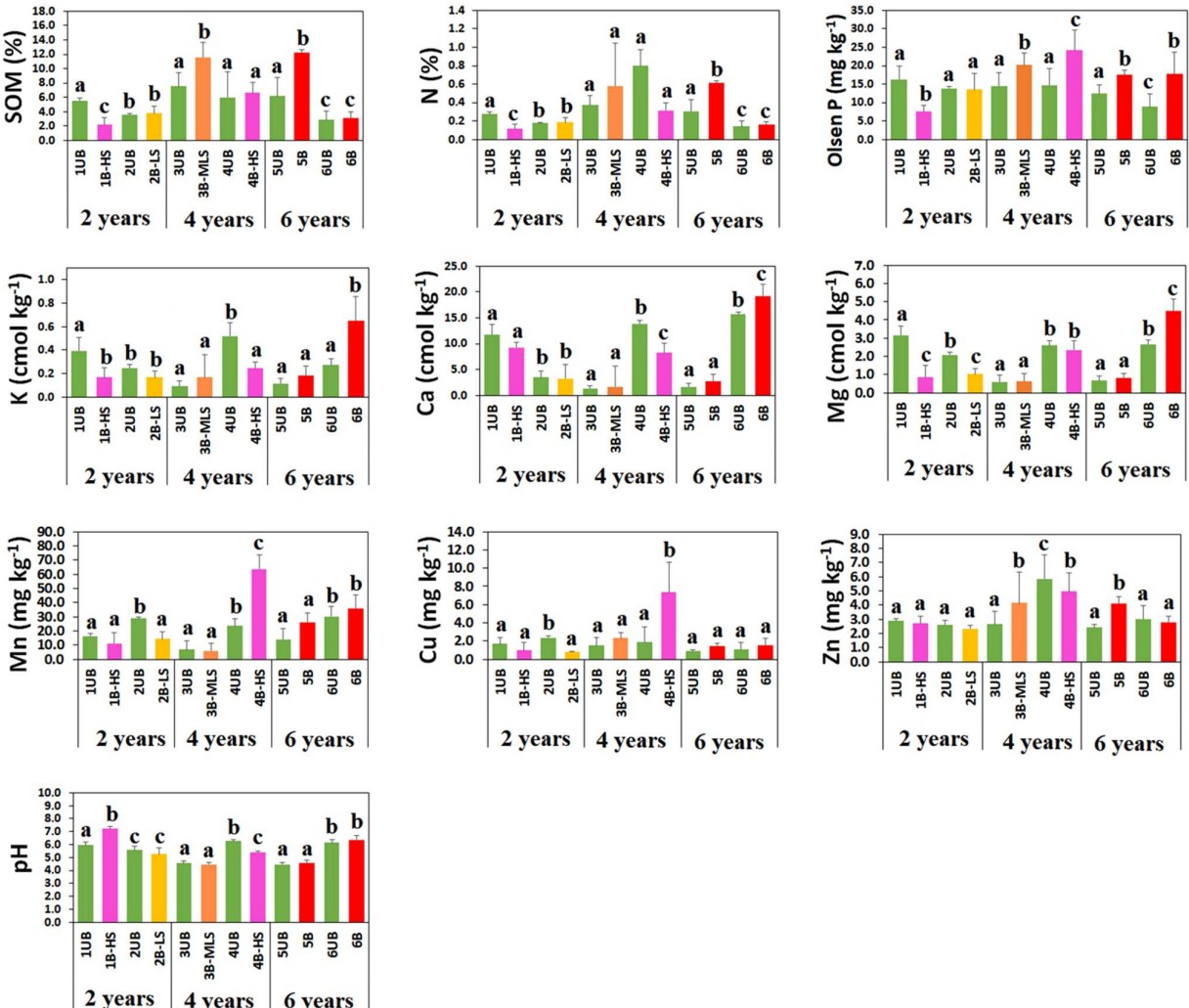

**Figure 5.** Effects of wildfires on organic matter (SOM), total N, P, K, Ca, Mg, Mn, Cu, Zn, and pH of the soils of the HMs of southern Ecuador. Results are presented for unburned areas (UB) with their respective burned area two (2019), four (2017), and six (2015) years after the fire. Furthermore, the degree of severity of each of the burned areas is presented. Bars stand for the standard error. Different letters mean significant difference among burning scenarios within each study year ($p < 0.05$, Tukey HSD).

### 3.3. Principal Components (PCA) of Soil Bulk Density and Nutrients, Applied to Burned HMs

Figure 6 presents the results of the PCA for the wildfires produced in the years 2019, 2017, and 2015. Component 1 explains 39.4% and component 2 explains 30.2% of the variance. In 2019, high severity burned (1B-HS) were negatively correlated with variables related to soil chemical fertility such as SOM, N, P, Cu, and Zn. In addition, it was positively related to bulk density (Bd) where there is greater soil compaction. This is conclusive with our results since we have found that, for a said year (2019), there is a decrease in these nutrients (Figure 5). The low severity fire of 2019 (2B-LS) does not correlate with nutrients.

In the case of the high severity fire (4B-HS) of 2017, it is positively correlated with SOM, N, P, Cu, Mn, and K. In addition, it was noted that the increase in these elements also occurred for the moderate-low severity fire (3B-MLS). Finally, for the 2015 wildfires, even though we do not know the degree of severity, it can be observed that there is a good correlation between SOM and N for the 5B fire, while for 6B, there is a good correlation with Mg, K, Ca, and pH. This suggests that in a time of approximately 4–5 years where studied HMs can maintain key functions and processes during the recovery of soil total nutrients.

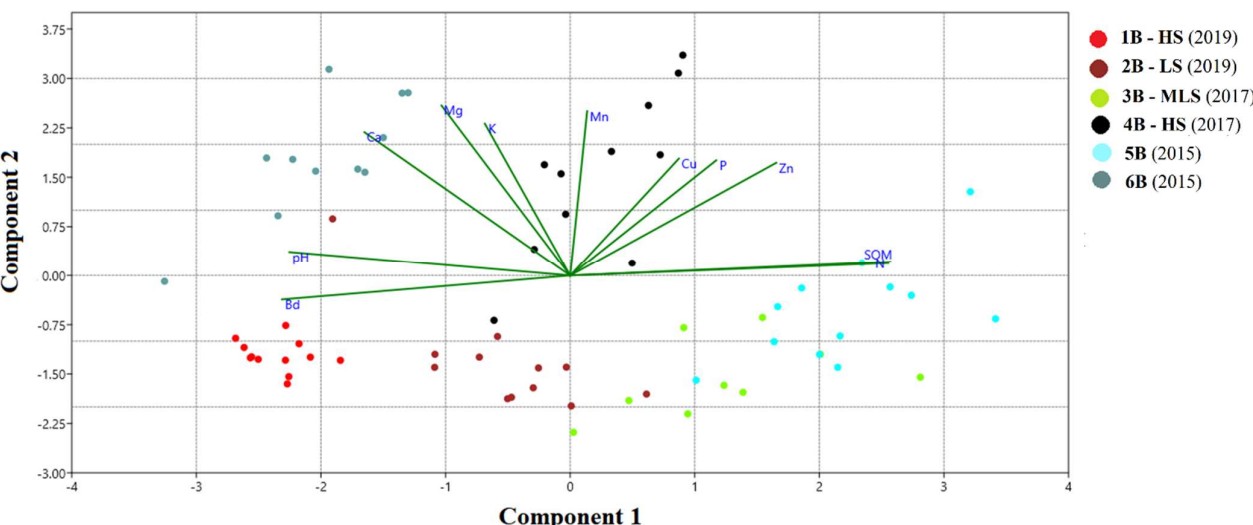

**Figure 6.** Orthogonal distribution of the Principal Component Analysis (PCA) of the parameters analyzed in burned HMs. The red circle represents the high severity fire of 2019 (1B-HS) and the brown circle the low severity fire of 2019 (2B-LS). The green circle is the moderate low severity fire of 2017 (3B-MLS) and the black circle is the high severity fire in 2017 (4B-HS). The bubble gum and lead-colored circles are the fires of the year 2015 where the severity is not known. Soil attributes include bulk density (Bd), organic matter (SOM); nitrogen (N); phosphorous (P); potassium (K), pH, calcium (Ca); magnesium (Mg), manganese (Mn), copper (Cu) and zinc (Zn).

## 4. Discussion

Wildfires in the studied HMs of southern Ecuador can occur during the period from July to October (4 months) when precipitation and relative humidity decrease and there is an increase in temperature and wind speed (Figures 3 and 4). This is consistent with the findings of Alves White [83] who, in his study in South America using remote sensing (hot spots detected using the MODIS sensor), determined that in most countries the key month for the occurrence of wildfires during the fire season (4 months) is September. However, the factor that determines the impact on soil physicochemical properties at a depth of 0 to 10 cm is the degree of severity, as has been shown in some research [84]. For example, Bielefeld et al. [85] determined that, in general, at this depth, fire directly affects soil physicochemical properties as they are exposed to surface heating [12]. This has been corroborated by works such as Chowdhury et al. [73] who found that a large part of the thermal energy generated during a fire is lost in the atmosphere, and a smaller amount is radiated downward and absorbed by organic matter. In this context, regarding the bulk density of the most recent and high severity fire (1B-HS: 2019, 2 years since the fire) (Table 2), it coincides with the results obtained by Goforth et al. [84] and Chandra and Bhardwaj [18] who showed that when fires are of high severity, the bulk density tends to increase in value and generates greater soil compaction. Likewise, Cerdà and Doerr [86] showed that the increase in bulk density is due to the collapse of aggregates and the clogging of pores by ash and dispersed clay minerals. As a consequence, soil porosity and permeability decrease as bulk density generally increases when ashes reach the first few centimeters of the soil [87]. In addition, increased bulk density soon after a wildfire has been associated with increased erosion rates and decreased soil SOM content [16]. In contrast, the low severity fire (2B-LS: 2019, 2 years after fire) led to a decrease in bulk density value and lower soil compaction compared to the unburned HM. This could be due to the higher amount of soil organic matter contributed by the vegetation that was not completely burned (low severity) that generally occurs in this type of fire [12,23].

On the other hand, with the passage of time (4 and 6 years after the fire), the natural regeneration processes of vegetation cause the bulk density to stabilize even though there was a high severity fire in 2017. This may be due to the fact that, as time passes,

there is a greater increase in SOM that allows such stabilization, as reaffirmed in recent research [18,88]. However, with respect to soil texture, where there is similarity in all the HMs studied, these results are not consistent with Ketterings et al. [89], who stated that soil texture is fundamentally affected when high intensity and severity burns occur. In the study area, there does not appear to be such an effect since it is known that, at higher temperatures, there is a reddening of the soil matrix, whereas in low to moderate intensity fires, the soil is covered by a layer of black ash or remains gray [87,90]. Perhaps for the HMs studied and for other ecosystems in southern Ecuador, this is a new topic for future research in which more thorough monitoring of changes in the sand, silt, and clay contents could be conducted.

Regarding the chemical properties of the soil, as expected in the most recent and high severity fire (1B-Hs of 2019), there was a loss of SOM with respect to 1UB, in contrast to the 2B-LS of 2019, where there was a relative increase in SOM, with respect to the 2UB (Figure 5). These results are consistent with those reported by Hrelja et al. [19] who indicated that high severity wildfires lead to more severe soil degradation, due to the burning of the SOM and the volatilization of some main nutrients such as C, N, and P [91]; while when fires of low thermal intensity and low severity (temperature reached and duration) occur, SOM and some total nutrients tend to increase over time, as reported by some recent research [12,17]. However, over time (4 years of fires in 2017), it is evident that even with high severity fires, SOM and some essential nutrients such as P, Mn, and Cu increase even with respect to HMs that never burned. This is consistent with that reported by Muqaddas [30], who for tropical humid forests reported that it takes 4 years for the recovery of SOM, C, and N contents. This same pattern is evident in the oldest fires of 2015 (6 years of recovery) where despite not knowing the level of severity, we suspect that 5B likely corresponded to a fire of low severity while zone 6B to a fire of low-moderate severity. This conclusion is supported by the work of Chandra et al. [18] and Certini [87] who found a significant increase in all nutrients (SOM, N, P, K, Ca, Mg, Mn, Cu, and Zn) in this type of fire (Figures 5 and 6). These results are consistent with what occurs in other ecosystems. For example, in the Pilbara biogeographic region (northwestern Australia), there is a leveling off of constant nutrient values 5 years after the fire, suggesting a partial regeneration of these ecosystems at that time [92]. Perhaps after 6 years in these ecosystems, there will also be a total recovery of the edaphic ecosystem as we have found in our study.

On the other hand, the pH range observed in Figure 5 (range 4.5–7.2) shows that this factor is in a range considered optimal for the soil [93]. For example, B, Cu, Mn, Ni, and Zn are known to be available over a pH range of 5.0 to 7.0, indicating that no limitation on their availability in the environment can be assumed for the study area. However, for other necessary nutrients (N, K, Ca, and Mg), these soil pH values are suboptimal since values between 6.5 and 8.0 are required [94]. Therefore, these results show the high resilience of these burned soils, at least for the studied physicochemical variables that could have recovered their pre-fire values, in about 6 years. In this context, it may be important to minimize SOM and nutrient losses by reducing soil erosion soon after the fire, especially in soils with lower pH and affected by high fire severity. In this regard, it is necessary to implement soil recovery activities in HMs in the first months after fire development in order to improve the pH value and minimize SOM and nutrient losses (avoid erosion). In this context, activities such as cutting burned trees (trunks), arranged following contour lines and fixed to the ground with stakes or stumps to prevent erosion are recommended [95]. These vegetable cordons are a very widespread post-fire action in the forestry world as a measure to control erosive processes (runoff and erosion). The implementations of these restoration practices, together with knowledge of fire severity and traditional knowledge of fire, are essential components for the formulation of new regulations and programs for the correct integrated management of fire in this anthropized ecosystem of southern Ecuador.

## 5. Conclusions

This study has identified the effects of wildfires on soil physicochemical properties in HMs of southern Ecuador affected during three contrasting periods (years 2019, 2017, and 2015). The results show that burned HMs present burn patterns of mixed severity ranging from high severity, moderate severity, and low severity. Negative effects on soil physicochemical properties occur in the most recent fires that are of high severity (the year 2019, 2 years since the fire), where soil nutrients are drastically lost and increased compaction (higher bulk density) occurs. However, over time (4 to 6 years after the fire), nutrients tend to increase and soil bulk density decreases and stabilizes, even compared to HMs that never burned, thus indicating that this ecosystem has a good resilience capacity due to natural succession processes. Consequently, to reduce the effects of wildfires, especially those of high severity in HMs, it is recommended to establish soil restoration practices in the first months after the burning of vegetation, since natural recovery would take 5 years. Furthermore, this type of restorative treatment must be initiated after the wildfire occurred. These findings can help decision makers to design policies, regulations, and proposals for the correct management and environmental restoration of the HMs of southern Ecuador affected by wildfires.

**Author Contributions:** Conceptualization, V.C.-P.; methodology, V.C.-P., M.B.H., L.J.Á., F.R.-B. and R.G.-R.; software, V.C.-P. and M.B.H.; validation, M.B.H., L.C.Q. and R.G.-R.; investigation, V.C.-P., M.B.H., L.J.Á., F.R.-B., L.C.Q. and R.G.-R.; resources, V.C.-P. and F.R.-B.; data curation, V.C.-P.; writing—original draft preparation, V.C.-P. and L.C.Q.; writing—review and editing, V.C.-P., M.B.H., L.J.Á. and R.G.-R.; visualization, V.C.-P.; supervision, R.G.-R.; project administration, V.C.-P. and F.R.-B.; funding acquisition, V.C.-P. and F.R.-B. All authors have read and agreed to the published version of the manuscript.

**Funding:** This research was funded by the UNIVERSIDAD TÉCNICA PARTICULAR DE LOJA, UTPL-PROY_VIN_AMB_2020_2706.

**Institutional Review Board Statement:** Not applicable.

**Informed Consent Statement:** Not applicable.

**Data Availability Statement:** Data are contained within the article or are available upon request.

**Acknowledgments:** Our thanks to the Universidad Técnica Particular de Loja for funding this research (UTPL-PROY_VIN_AMB_2020_2706) and to Gregory Gedeon for reviewing the English text. We acknowledge the use of data and/or imagery from NASA's Fire Information for Resource Management System (FIRMS) (https://earthdata.nasa.gov/firms, accessed on 15 April 2021), part of NASA's Earth Observing System Data and Information System (EOSDIS).

**Conflicts of Interest:** The authors declare no conflict of interest. The funders had no role in the study design; in the collection, analysis, or interpretation of data; in the writing of the manuscript; or in the decision to publish the results.

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
