# Peer review of "Effects of the Severity of Wildfires on Some Physical-Chemical Soil Properties in a Humid Montane Scrublands Ecosystem in Southern Ecuador"

_fire, doi:10.3390/fire5030066_

Round 1

Reviewer 1 Report

While the topic and analyses are important to understanding the effect of fire on soil properties globally, this manuscript is not ready for publication. Please refer to the attached Review Report.

Author Response

Manuscript ID: fire-1648878

Dear Editors,

We appreciate the constructive comments that the editor and reviewers have devoted to our manuscript.

Attached to this cover letter, you will find the responses to the reviewers’ comments highlighted in a different color "green". In brief, we have taken into account all the reviewer's concerns and comments and have modified the required parts accordingly.

We hope that the changes have improved the quality of our manuscript and that it can now be considered for publication in the journal Fire.

We look forward to your decision.

Sincerely yours,

Dr. Vinicio Carrión-Paladines.

___________________________________________________________________________

REVIEWER: 1

  1. Brief summary:

The aim of the manuscript was to present data and analysis of fire-induced changes (or lack thereof) to soil physio-chemical properties in a scrubland system in southern Ecuador. Comparable control (unburned) and treated (burned) sample sites were established near and within burned areas of different ages. The Normalized Burn Ratio (NBR) index was used to characterize fire sites. Analysis of Variance (ANOVA) and Principal Component Analysis (PCA) were used to analyze and characterize salient features of the data. The application of the analysis to local management was briefly noted. 

It is generally acknowledged that the occurrence and size of wildfires (whether natural or man-made) are expected to increase under current and projected climate scenarios. Understanding ecosystem effects of wildfire is important to understanding the health, functioning, and management of the system. Specifically, the data and analysis presented in the reviewed manuscript are a necessary part of the growing understanding of the response of soil—a critical component of the ecosystem—to wildfire.

Initially, it appeared the manuscript would require a simple English language review. However, upon further reading, it was evident there are serious flaws in the manuscript. These flaws include: inconsistent, even improper, use of terms; unclear data presented in the figures and tables; and vague reasoning leading to questionable inferences.

Thank you for your comment; however, we have considered all of the reviewers’ comments and have made all necessary corrections. Therefore, we ask for you to take them into account.

  1. General comments

As necessary revisions to the manuscript would likely require more than the ten-day revision period allowed by this journal, the overall recommendation is for rejection of this manuscript for publication at this time.  

Thank you for your comment; however, we have made all necessary corrections requested by the reviewer, within the time frame established by the journal. Therefore, we ask for you to consider them.

  1. Specific comments:

(Line 3)           After a brief search on the internet, it seems the more common usage for “scrub ecosystem” is “scrubland ecosystem” in the title. Therefore, reference would be to “scrublands” rather than “scrubs” throughout the manuscript.

Thank you for your comment. We have replaced the term “scrubs” with “scrublands” throughout the manuscript.

(Line 23)         NBR is not the normalized burn area index but, rather, the normalized burn ratio (NBR), an areal index.

This correction is much appreciated. We are very sorry for the mistake. We completely agree that NBR is Normalized Burn Ratio. It has been corrected.

(Line 23)         The term “differential” usually refers to the derivative; the word “difference” is more appropriate throughout the manuscript.

Thank you... we agree with your comment; therefore, we have replaced the term “differential” with “difference” throughout the manuscript.

(Line 34)         The phrase “edaphic properties” was listed as a keyword, but “edaphic” was used only once (apparently) and without definition in the manuscript. Substituting “soil properties” would be more representative.

Thank you for your comment. We have included the phrase "soil properties" throughout the manuscript for consistency throughout the manuscript text.

(Line 157)       The coordinates for the location of Loja canton are not meaningful in the opening sentence of the study area description.

Thank you for your comment. The respective adjustment has been made.

(Line 176, Figure 1)   This figure is confusing and not helpful to readers unfamiliar with the area. The relationship between inset boxes and boxes in Figure 1 is not clear. The figure in the inset box does not seem to match the presumably enlarged figure to the right. All fonts in the figure are too small, virtually illegible.

Thanks for your comment... we have constructed a new figure considering all of the reviewers’ recommendations.

(Lines 184-190)          This section seems to be incomplete (“citation” to what?) and should be rewritten, possibly presented as a table.

Thank you for your comment. We have incorporated the bibliographic citations correctly and have rewritten the paragraph for better understanding.

(Line 205, Table 1)     The table title seems more a list of characteristics of the HMs, as specific coordinates and elevations are given. In the header, “Zones” are listed but there is no description of a zone in the text. The next header to the right refers to time to chemical analysis only. Perhaps “time” could be simplified to “time after fire”, e.g., Year 2, Year 3, Year 4 etc. The next header seems more indicative of the type of treatment (unburned or burned) rather than MHs. The coordinates listed in the next column end with O rather than W. Altitude is listed as the next header although Elevation was discussed in the text. This type of table (and related text) would be helpful as a way to characterize the experimental design and, especially, the sample naming convention.

All these comments are much appreciated. We have tried to clarify it in the text and in Table 1.

(Line 211)       The term “climograms” is not a standard term. The words “climatogram” and “climagraph” are more appropriate, with “climagraph” being the standard term.

Thank you for your comment. In this context, we have changed the term "climagraph" to "climograms" throughout the text of the manuscript as indicated by the reviewer.

(Line 237- paragraph)            The explanation of the sampling is confusing. For each different treatment (site?, burn treatment?, year? there are four replicates? This is very important information to make clear for the study.

            Also, the type of soil analyses included physical properties (texture, bulk density) and “chemical” properties although only pH, SOM, and soil nutrients were studied. It would be helpful to indicate this clearly and early (possibly in the abstract). Also, it is stated in line 253 that soil was sampled for iron. The results are not presented in Tables 3 or 4 nor discussed.

We have clarified it as follows “In each of the study sites, under burned and unburned treatments, three 20 m x 20 m plots (400 m2 each, 2400 m2 per site, 14400 m2 total) were installed. Each plot was evenly spaced at least 100 m apart. Soil samples were taken at each corner of each plot at a depth of 0 - 10 cm, using standardized metal cores (6 cm diameter, 10 cm height, 283 cm3 volume) [69]. In each plot, 4 samples were taken separately for further analysis of bulk density, texture, pH, soil organic matter (SOM), and total soil nutrients (3 years * 2 study sites * 2 treatments * 3 plots x 4 samples=144 individual samples). Once the soil samples were collected, they were packaged in separately labeled polypropylene bags."

 (Line 260)      Tukey’s honestly significant difference (HSD), a post hoc test, was discussed later in the manuscript.

It has been correctly specified in the text.

(Line 270)       Climagraphs and burn severity apply more to site characterization than to the Results and discussion section.

We agree with the reviewer's suggestion. Now it has been moved to the material and method section.

(Line 320, Figure 2)   This Figure needs work. Again, all fonts (except for the Legend) are too small. The relative humidity and windspeed data on the climographs have no numerical axes. The burn severity maps are not easily related to the specific sampling sites. Perhaps the 20m x 20m sampling plots, and sampling points within each plot, could be indicated on the burn severity plots, or else a general sampling plot and point figure could be generated.

Thank you for your comment. The respective adjustments have been made to the climographs by placing each variable in color and increasing the font size on each of the axes. In addition, the relative humidity and wind speed have been placed on the numerical axes in the climographs. As for the burn severity maps, two images were redrawn and the plots of burned sites were incorporated into the map. We believe that both climagrophs and severity maps are now described in greater detail, based on the reviewer's suggestion.

(Line 270 – Section 3.1)        The discussion of climographs and burn severity is unnecessarily long, can be summarized, and belongs in Section 2 (Materials and Methods) as neither data set is a new result nor the objective of the paper. Additionally, characterization of the sampling data by the burn severity maps seems arbitrary. The degree of burn severity over an area is not the same as the burn severity at the points at which the samples were taken. The burn severity would belong more in characterization of the sample location and experimental design. The effect of burn severity on experimental results, however, does belong in the Results section.

We agree with the reviewer's suggestion. Therefore, only the relevant information on climate and fire severity in the study plots has been reported in the material and methods section.

(Line 360 – Section 3.2) The discussion of results is complicated by the sudden introduction of sample names without presenting the system of how samples are identified. The discussion here refers to sample names and Figure 2 but there is no connection to the sample names in Figure 2.

To clarify, the plot codes are now described in table 1 and used consistently along the whole text, figures, and tables.

(Line 394 – Table 2) The designation of the samples needs to be explained; it is helpful to the reader who is unfamiliar with the data. Specifically, the abbreviations should be identified in the caption. Sometimes, the names given to field samples and analyses need to be clarified and simplified for publication. If all the samples are in HMSs, then it is not significant as a sample designation.

For clarity, the plot codes are now described in table 1 and used consistently along with the whole text, figures, and tables.

(Line 438 – Figure 3) All fonts are too small. The year designation for the graphs could be simplified as Year 2, Year 4, etc.  Also, the term “macronutrients” is introduced in the caption and should have been described in the text, for example, in the methods section.

We have increased the font size in the figure and they are now also colored for better understanding. For clarification, plot codes are now described in Table 1 and are used consistently throughout the text, figures, and tables. To avoid any misunderstanding, we have used only "nutrients" throughout the text.

(Line 460 – Figure 4) All fonts are too small. The term “micronutrients” is introduced in the caption but not introduced in the text. Ca and Mg are normally considered macronutrients.

(See Hossner L.R. (2008) Macronutrients. In: Chesworth W. (eds) Encyclopedia of Soil Science. Encyclopedia of Earth Sciences Series. Springer, Dordrecht. https://doi.org/10.1007/978-1-4020-3995-9_337)

Thank you for your comment. The figure has been improved by increasing the font size and presenting it in color for better understanding. In addition, to avoid any misunderstanding, the term "nutrients" has been used throughout the text.

(Line 513)       The section “Author Contributions” was not completed.

Thank you for your comment. The information regarding author contributions has been completed.

Reviewer 2 Report

 Abstract

I cant see any quantitative results in this part. So, this is important to write this part again.

  1. Introduction

What is innovative of the current study?

2.1. Study Area

Please write coordinate systems of the study area in DMS (Degree, Minutes, Seconds)

  1. Materials and Methods

Please add a flowchart for the current study.

Please add equation numbers.

2.4. Soil sampling and laboratory analysis

How do you select these soil samples?

Why 0-10cm? what is your reason for this sampling?

  1. Results and discussion

I am not satisfied from this part.

Please explain your achieved results, then concentrate on discussion. This is important to separate these parts.

In total, discussion is written poor.

Author Response

Manuscript ID: fire-1648878

Dear Norah Wang

We appreciate the constructive comments that the editor and reviewers have devoted to our manuscript.

Attached to this cover letter, you will find the responses to the reviewers’ comments highlighted in a different color "green". In brief, we have taken into account all the reviewer's concerns and comments and have modified the required parts accordingly.

We hope that the changes have improved the quality of our manuscript and that it can now be considered for publication in the journal Fire.

We look forward to your decision.

Sincerely yours,

Dr. Vinicio Carrión-Paladines.

___________________________________________________________________________

REVIEWER: 2

 Abstract

I cant see any quantitative results in this part. So, this is important to write this part again.

Thank you for your comment. We have included in the abstract the values of the PCA (components 1 and 2) and ANOVA so that there is a better understanding of the results.

  1. Introduction

What is innovative of the current study?

Thank you for your comments. We have included in the introduction the importance of this research (you can review it on lines 119 - 123).

2.1. Study Area

Please write coordinate systems of the study area in DMS (Degree, Minutes, Seconds)

Thank you for your comment. We have taken into account suggestions from other reviewers who recommended placing them in decimal degrees. We have also made a new map of the study area including decimal degrees.

  1. Materials and Methods

Please add a flowchart for the current study.

Thank you for your comment. We have included in the materials and methods section the flow diagram of the study, as requested by the reviewer.

Please add equation numbers.

Thank you for your comments. We have added respective numbers in the equations.

2.4. Soil sampling and laboratory analysis

How do you select these soil samples?

Thank you for your comment. This part has been rewritten indicating in the methodology section how the samples are selected.

Why 0-10cm? what is your reason for this sampling?

Thank you for your comment. Also in the methodology section we include the reasons why the 0 - 10 cm depth was used. It is now clearer.

  1. Results and discussion

I am not satisfied from this part.

Please explain your achieved results, then concentrate on discussion. This is important to separate these parts.

In total, discussion is written poor.

Thank you for your comment. We have separated the results and the discussion and deepened the discussion by including new bibliographic citations so that it is now more understandable and within the format of the journal.

Reviewer 3 Report

NBR index is not really a measure of burned severity. It is based on the proportion of the vegetation that remained unburned as compared to the proportion of the area where the vegetation is burned. Burn severity is classified as 'white ash' (severe burn), 'bare soil' (moderate burn) and ' black soil' (light burn) conditions. According to the burn intensity, certain changes in the physico-chemical properties of the soils have been extensively documented in the literature. The results of this study do not coincide with the established literature (i.e., Nitrogen, Phosphorus and pH values in 2019 and 2017). Also, the results are contradictory to each other from year to year; SOM and pH between years 2015 and 2019.

Overall, the amount of nitrogen in the soil seems to be very low in all years measured. Phosphorus in 2019 is less in the severely burned site than in the unburned site. According to the existing literature, this is an error of fact.

The soil slope is 70%. This is extremely steep. Given the torrential nature of rains in that part of the world, loss of nutrients with soil erosion might be an explanation of the dubious results.

The scale of values on the y-axis of all three diagrams in figure 3 is different. This makes it very difficult to compare each value among the three years studied. 

SM

Author Response

Manuscript ID: fire-1648878

Dear Norah Wang

We appreciate the constructive comments that the editor and reviewers have devoted to our manuscript.

Attached to this cover letter, you will find the responses to the reviewers’ comments highlighted in a different color "green". In brief, we have taken into account all the reviewer's concerns and comments and have modified the required parts accordingly.

We hope that the changes have improved the quality of our manuscript and that it can now be considered for publication in the journal Fire.

We look forward to your decision.

Sincerely yours,

Dr. Vinicio Carrión-Paladines.

___________________________________________________________________________

REVIEWER: 3

NBR index is not really a measure of burned severity. It is based on the proportion of the vegetation that remained unburned as compared to the proportion of the area where the vegetation is burned. Burn severity is classified as 'white ash' (severe burn), 'bare soil' (moderate burn) and ' black soil' (light burn) conditions.

We appreciate your comment. However, when we analyze the evolution of vegetation in a burned area, we often do not have information on the fire severity of the vegetation immediately after the fire event. The application of supervised techniques (classification, regression, etc.), which requires knowing the fire severity in a certain number of plots in the area, is complicated when we try to study fire events of past years, for which the field information obtained in the weeks after the fire is scarce or non-existent, and even more so when the period is long as in the case of this study (2, 4 and 6 years). Despite their limitation, the usefulness of post-fire NDVI and NBR, as well as the respective bi-temporal indices (difference between the index value before and after the fire), dNDVI and dNBR, to discriminate burned areas and fire severity, has been proven in previous studies (e.g., Escuin et al 2008; Miller and Thode 2007; Murphy et al., 2008).

According to the burn intensity, certain changes in the physico-chemical properties of the soils have been extensively documented in the literature. The results of this study do not coincide with the established literature (i.e., Nitrogen, Phosphorus and pH values in 2019 and 2017). Also, the results are contradictory to each other from year to year; SOM and pH between years 2015 and 2019. Overall, the amount of nitrogen in the soil seems to be very low in all years measured.

In our study, the fire severity data (dNBR) might not match the effect described in the literature, among other factors because we sampled a few years after fire occurrence and factors other than fire might affect soil properties (e.g. post-fire erosion, post-fire plant establishment, etc.). This is explained for example by the work of Escuin, S., Navarro, R., & Fernandez, P. (2008). Fire severity was assessed using NBR (Normalized Burn Ratio) and NDVI (Normalized Difference Vegetation Index), derived from LANDSAT TM/ETM imagery. International Journal of Remote Sensing, 29(4), 1053-1073.

In addition, the conditions between sectors, although similar, have characteristics that may differ, due to the set of interacting factors that could cause these differences between years (climograhs, fire severity in the sampled sites). The average values of nitrogen for 2017 and 2019 were 0.45 and 0.15 respectively and in the case of pH presented averages of 4.95 and 6.25 respectively. A similar case occurred when comparing the burned areas of the years 2015 - 2019, when the average values of organic matter were 7.7 % older fire "year 2015" and 3 % more recent fire in "year 2019" respectively. This is possibly due to natural regeneration processes and the effect of time (years) in the cycling of nutrients. As for nitrogen, the values presented were 0.4 and 0.15 % respectively, indicating that the recent fires could have burned most of the residues in the superficial part of the soil and there could have been losses by volatilization.

The same is true for pH. When averaging the soil pH in the burned areas, the highest value is found in 2019 (6.25) compared to 2015 (5.5), which shows that burning increases the soil pH due to the contribution of ash. In short, burned areas have different topography characteristics, so the impact of fires can be extremely variable, where after burns or fires there can be degenerative dynamics due to rainfall, such as runoff, leaching, and erosion (Lopez 2006; Chowdhury et al., 2022).

Phosphorus in 2019 is less in the severely burned site than in the unburned site. According to the existing literature, this is an error of fact.

We appreciate your comment. However, the results obtained for phosphorus (lower phosphorus content in the 2019 high-severity burned site versus the unburned site) are consistent with other recently published studies. For example, Kelly et al. (2021) showed that when a fire is of high severity, it causes a substantial loss of SOM and volatilization losses of nitrogen, phosphorus, and potassium. Other studies such as Hebel et al., (2009) reaffirm that high severity wildfires lead to more severe soil degradation, such as the volatilization of some of the main soil nutrients, namely carbon, nitrogen, phosphorus, and sulfur. In short, the existing literature is consistent with our findings.

The soil slope is 70%. This is extremely steep. Given the torrential nature of rains in that part of the world, loss of nutrients with soil erosion might be an explanation of the dubious results.

Thanks for your comment. However, the results are not doubtful, rather they are consistent since a fire of high severity and on this type of slope (70%), also causes an alteration in the stability of the soil aggregates, which can cause, depending on climatic conditions (high rainfall), erosion and runoff after forest fires, as shown by some recent studies (e.g. Pereira et al. 2018). This comment is much appreciated.

The scale of values on the y-axis of all three diagrams in figure 3 is different. This makes it very difficult to compare each value among the three years studied. 

SM

The scale of values on the y-axis of all three diagrams in figure 3 has been changed and now it is the same in all cases.

Reviewer 4 Report

Effects of the severity of wildfires on some physical-chemical soil properties, in a humid montane scrub ecosystem in southern Ecuador

by Vinicio Carrión-Paladines, M. Belén Hinojosa, Leticia Jiménez Álvarez, Fabián Reyes-Bueno, Liliana Correa Quezada and Roberto García-Ruiz

In this manuscript, the authors present the results of a well-executed study of the effects of wildfires on physical and chemical parameters of soil in the Humid montane scrubs ofsouthern Ecuador. The authors analyze the influence of fire severity on soil properties using remote sensing data as well as laboratory analysis of soil samples. The study demonstrates that the fire severity in humid montane scrubs can vary significantly determining negative effects of fires.

In my opinion, the topic is of interest from both scientific and practical points of view. I believe that the manuscript is within the scope of the Fire journal and can be accepted for publication after revision.

Major concerns:

  1. Why do the authors perform the analysis of soil characteristics only from a part of the soil profile (0–10 cm)? The authors also do not provide the morphological characteristics of the soil profile, which does not allow assessing which horizons of the soil profile correspond to this part of the soil profile. Can the results of this analysis be applied to the whole soil profile or should they be differentiated by layers (depths). Please discuss this issue in more detail.
  2. Soils in the considered plots differ significantly in acidity (pH), which can significantly affect the mobility of elements.The manuscript provides only total concentrations of elements. I think it would be reasonable provide also the mobile forms of the elements, since pyrogenic impact could significantly change the number of mobile forms.

Minor concerns:

  1. Lines 157 – 158: Consider using decimal degrees for specifying study area location. The same remark is for Figure 1 as well – either to use decimal degrees or at least provide coordinate system information.
  2. Line 175: “The predominant soils in HMs are Entisols and Inceptisols.” Is it possible to use the international classification of soil types according to the World Reference Base for Soil Resources (WRB*)?
    *World Reference Base for Soil Resources 2014, Update 2015. International soil classification system for naming soils and creating legends for soil maps. 3rd. — Rome : FAO, 2015. — ISBN 978-92-5-108370-3.
  3. Lines 182–235: The authors do not provide information about the fires where soil samples were taken. I think it would be reasonable to show, for instance, as contours of fire polygons in Fig. 2. Consider also providing information on fire dates, burned areas, types of vegetation cover, perhaps as an additional figure.
  4. Lines 183 – 190: Why do the authors use MODIS/VIIRS active fire products but not burned area products (MCD64A1/ VNP64A1)?
  5. Lines 202 – 203: Please check the altitude range in the following statement “…and altitudes (from 1660 203 to 2320 m asl)”, since in Table 1 (4/2017) max altitude is 2658 m asl.
  6. Table 1: What does the letter “O” means in the “Coordinates” column? Maybe it should be changed to “W” for western longitude?
  7. Lines 214 – 215: Is it possible to use other data, for instance Landsat for the year 2015?
  8. Lines 217 – 218: Since the authors use NBR to highlight burned areas, I cannot figure out what is the reason to use also active fire data from MODIS and VIIRS.
  9. Line 217, 326: NBR actually stands for “normalized burn ratio”, not for “normalized burned area”.
  10. Figure 2: Axes of the climograms in Fig. 2 show units only for precipitation and temperature, but not for wind speed and humidity. Also, a reference is required for the severity classification scheme (lower right corner of Fig.2). If this classification is taken from [67], [68], then I think it is necessary to discuss the possibility of their use for the considered study area.
  11. There are at least 4 fire severity classes in Fig. 2.However, in the results the authors consider only High severity, Low severity and Moderate-low severity (fig 3, 4, 5, table 2).Please explain why not all severity classes are considered.
  12. Lines 323 – 325: Please check zone numbers in Fig. 2 and Table 1. For the year 2017 zones are numbered as 3 and 4, but not 1 and 2. The same is for the year 2015.
  13. Table 2, Figure 3 and 4: I cannot figure out how letters (a, b, c) are related to mean significant difference.
  14. Figure 5: If fires of the year 2015 were not classified by severity, then why points on the principal component plot are distributed in the opposite directions? Please discuss this issue in more details.

Author Response

Manuscript ID: fire-1648878

Dear Norah Wang

We appreciate the constructive comments that the editor and reviewers have devoted to our manuscript.

Attached to this cover letter, you will find the responses to the reviewers’ comments highlighted in a different color "green". In brief, we have taken into account all the reviewer's concerns and comments and have modified the required parts accordingly.

We hope that the changes have improved the quality of our manuscript and that it can now be considered for publication in the journal Fire.

We look forward to your decision.

Sincerely yours,

Dr. Vinicio Carrión-Paladines.

___________________________________________________________________________

REVIEWER: 4

In this manuscript, the authors present the results of a well-executed study of the effects of wildfires on physical and chemical parameters of soil in the Humid montane scrubs ofsouthern Ecuador. The authors analyze the influence of fire severity on soil properties using remote sensing data as well as laboratory analysis of soil samples. The study demonstrates that the fire severity in humid montane scrubs can vary significantly determining negative effects of fires.

In my opinion, the topic is of interest from both scientific and practical points of view. I believe that the manuscript is within the scope of the Fire journal and can be accepted for publication after revision.

 Thank you for your positive comment. All the revisions raised by you have been duly developed.

Major concerns:

  1. Why do the authors perform the analysis of soil characteristics only from a part of the soil profile (0–10 cm)? The authors also do not provide the morphological characteristics of the soil profile, which does not allow assessing which horizons of the soil profile correspond to this part of the soil profile. Can the results of this analysis be applied to the whole soil profile or should they be differentiated by layers (depths). Please discuss this issue in more detail.

Thank you for your comment. The study was carried out in the superficial layer of the soil (0 - 10 cm) because fire generally increases the temperature at this depth and mainly affects the edaphic properties since they are directly exposed to surface heating. The effect of fire decreases as the depth of the soil increases. Much of the thermal energy released is lost in the atmosphere and a smaller amount is radiated downwards and absorbed by the organic part, as previous studies have shown (DeBano; 1990). What's more, the works of Chowdhury et al., (2022); Bielefeld and da Cunha, (2003) were also carried out in the first 10 centimeters of the soil (from 0 to 5 and from 0 to 10 cm), so we consider this depth based on the previous important studies. In addition, we have included in the discussion section this information in the text of the manuscript.

  1. Soils in the considered plots differ significantly in acidity (pH), which can significantly affect the mobility of elements.The manuscript provides only total concentrations of elements. I think it would be reasonable provide also the mobile forms of the elements, since pyrogenic impact could significantly change the number of mobile forms.

Thank you for your comment. We agree with what the reviewer suggests, as pH affects the mobile forms of the elements; however, this is relevant especially when the land use pertains to cultivated areas. In this case, we have considered the total concentrations of the elements based on some previous studies (e.g. Verma et al. 2019; Li et al. 2020), and also since the vegetation of the studied areas is mainly scrubland, the same ones that are adapted to the conditions of the area. For future studies we consider performing the analyses of mobile forms in the laboratory, so we thank the reviewer for this suggestion.

Verma, S., Singh, D., Singh, A. K., & Jayakumar, S. (2019). Post-fire soil nutrient dynamics in a tropical dry deciduous forest of Western Ghats, India.

Li, X., Jin, H., Wang, H., Wu, X., Huang, Y., He, R., ... & Jin, X. (2020). Distributive features of soil carbon and nutrients in permafrost regions affected by forest fires in northern Da Xing’anling (Hinggan) Mountains, NE China. Catena185, 104304.

Minor concerns:

  1. Lines 157 – 158: Consider using decimal degrees for specifying study area location. The same remark is for Figure 1 as well – either to use decimal degrees or at least provide coordinate system information.

Thank you for your comment. We have adjusted the use of decimal degrees and also improved Figure 1 for better understanding.

  1. Line 175: “The predominant soils in HMs are Entisols and Inceptisols.” Is it possible to use the international classification of soil types according to the World Reference Base for Soil Resources (WRB*)?
    *World Reference Base for Soil Resources 2014, Update 2015. International soil classification system for naming soils and creating legends for soil maps. 3rd. — Rome : FAO, 2015. — ISBN 978-92-5-108370-3.

We thank you for your positive feedback. We have replaced the bibliographic citation considering the reviewer's suggestion: 61. IUSS Working Group WRB. (2015). International soil classification system for naming soils and creating legends for soil maps. World reference base for soil resources 2014, update 2015, 106.

  1. Lines 182–235: The authors do not provide information about the fires where soil samples were taken. I think it would be reasonable to show, for instance, as contours of fire polygons in Fig. 2. Consider also providing information on fire dates, burned areas, types of vegetation cover, perhaps as an additional figure.

Thank you for your comment. Figure 2 has been adjusted, contours have been added, and in addition the dates of fires as suggested by the reviewer.

  1. Lines 183 – 190: Why do the authors use MODIS/VIIRS active fire products but not burned area products (MCD64A1/ VNP64A1)?

Thank you for your comment and we agree with the reviewer. However, we consider that with all three products the burned sites can be identified. We use MODIS/VIIRS only to identify fire hotspots, which would then be confirmed and delimited with Sentinel 2 for 2017 and 2019.

  1. Lines 202 – 203: Please check the altitude range in the following statement “…and altitudes (from 1660 203 to 2320 m asl)”, since in Table 1 (4/2017) max altitude is 2658 m asl.

Thank you for your comment and we have indeed corrected the maximum altitude of 2658 m asl.

  1. Table 1: What does the letter “O” means in the “Coordinates” column? Maybe it should be changed to “W” for western longitude?

Thank you for your comment. We have indeed corrected the error. Now the longitude is in "W" in the table.

  1. Lines 214 – 215: Is it possible to use other data, for instance Landsat for the year 2015?

Thank you for your comment, however, we found a cloud-free Landsat image from a date after the 2015 fires which could only be identified 8 to 10 months after the fire, after the rainy period (austral winter). Therefore, it was not reliable to calculate the NBR with this data because it was too late.

  1. Lines 217 – 218: Since the authors use NBR to highlight burned areas, I cannot figure out what is the reason to use also active fire data from MODIS and VIIRS.

Thank you for your comment. We used MODIS/VIIRS only to identify fire hotspots, which would then be confirmed and delineated with Sentinel 2 for 2017 and 2019.

  1. Line 217, 326: NBR actually stands for “normalized burn ratio”, not for “normalized burned area”.

Thank you for your comment.... we have adjusted the terminology as suggested by the reviewer.

  1. Figure 2: Axes of the climograms in Fig. 2 show units only for precipitation and temperature, but not for wind speed and humidity. Also, a reference is required for the severity classification scheme (lower right corner of Fig.2). If this classification is taken from [67], [68], then I think it is necessary to discuss the possibility of their use for the considered study area.

Thank you for your comment. We have re-constructed the climographs in full color and included the units for both precipitation (mm), relative humidity (%), wind speed (km/h), and temperatura (°C). We have also re-constructed the severity maps including the location of the plots in the burned HMs and the severity classification as suggested by the reviewer. In addition, we considered the use of the classification of Parker et al. (2015) and Santos et al. (2020).

  1. There are at least 4 fire severity classes in Fig. 2. However, in the results the authors consider only High severity, Low severity and Moderate-low severity (fig 3, 4, 5, table 2). Please explain why not all severity classes are considered.

Thank you for your comment. To clarify the severity types in each study area, we have enlarged the severity maps and placed the sampling plots. They are now clearer.

  1. Lines 323 – 325: Please check zone numbers in Fig. 2 and Table 1. For the year 2017 zones are numbered as 3 and 4, but not 1 and 2. The same is for the year 2015.

Thank you for your comment. As we have improved both climograhs and severity maps, the respective numbers have also been checked.

  1. Table 2, Figure 3 and 4: I cannot figure out how letters (a, b, c) are related to mean significant difference.

Thank you for your comment. An explanation has been placed in Table 2 and figures: Different letters mean a significant difference (p < 0.05, Tukey HSD).

  1. Figure 5: If fires of the year 2015 were not classified by severity, then why points on the principal component plot are distributed in the opposite directions? Please discuss this issue in more details.

Thank you for your comment. We have rewritten indicating that for the 2015 year HMs B1 possibly corresponds to a low severity fire as reported by Chandra et al. 2015 and HMs B2 possibly corresponds to a moderate low severity fire as reported by Certini (2005).

Round 2

Reviewer 1 Report

The responsiveness and effectiveness of the authors' response to the first review is commendable. Following are a few comments:

Line 19 (Abstract):  The abstract is clarified. Specifically, the results of the PCA are simply and clearly articulated. Unfortunately, these clear statements are not reflected in the conclusion. 

Line 181:  Figure 1 is much improved and informative!

Line 190:  Figure 2 could be simplified. A return arrow from Step 3 to Step 2 is shown and not discussed in the text. 

Line 210:  Table 1 should be placed after the discussion about how the attributes of the sampling were identified--i.e., after the discussion of burn severity (at the end of section 2.4). 

LIne 228:  The climographs each show 4 different types of data (precipitation, temperature, relative humidity, and wind speed) but only two y-axes are shown on the graphs. The graphs are not readable without axes labels and the missing additional axes. Why are there two climographs for 2017? Also, as the climographs are being used to characterize meteorological parameters each year, the axes ranges for the particular data set should be the same. It is not possible to read information from the annual climographs if the axes ranges are not the same, and if the data values are not associated with an axis.

Line 260:  Figure 4 is improved. However, characterization of the fire severity of the map scene (as shown) is incorrect...the map scene is not the fire and characterization of the fire severity of the map scene depends only on which particular scene is chosen. However, showing the location of a specific sampling site at a location characterized by burn severity is helpful. 

Lines 366 and 371:  Figures 5 and 6 are basically incomprehensible. Again, quantitative data are graphed without axes and units. These two graphs can combined by simply showing each soil parameter in a row by year with appropriate axes in the same ranges. Please do not make the reader interpret the analyses.

Line 399:  Section 4.1 is not necessary. The climograph analysis is not necessary and can be summarized as identifying the fire season in a couple sentences. It is well-established that burn severity is not consistent throughout a fire. The discussion of the burn severity patterns is meaningful if analyzing the entire fire rather than the map scenes shown in the manuscript, which are arbitrary boundaries. There are several analysis tools to describe the attributes of the burn severity pattern for a fire...but that is not an objective of this manuscript.

Line 596:  Greater than 100 references for this research paper, the focus of which is data from a relatively unstudied area, seems excessive.

Author Response

Manuscript ID: fire-1648878

Dear Norah Wang

We appreciate the constructive comments that the editor and reviewers have devoted to our manuscript.

Attached to this cover letter, you will find the responses to the comments of reviewers 1 and 3 highlighted in another "green" color. In summary, we have taken into account all the reviewers' concerns and comments and have modified the required parts accordingly.

We hope that the changes made have improved the quality of our manuscript and that it can now be considered for publication in the journal Fire.

We look forward to your decision.

Sincerely yours,

Dr. Vinicio Carrión-Paladines.

Reviewer 3 Report

Considering the explanations you provided, I have a more clear picture of what you did and why. The paper is improved although its scientific merit remains average.

Author Response

(The authors gave the same response as above.)
